# The Pathology of Parkinson’s Disease and Potential Benefit of Dietary Polyphenols

**DOI:** 10.3390/molecules25194382

**Published:** 2020-09-24

**Authors:** Sunisha Aryal, Taylor Skinner, Bronwyn Bridges, John T. Weber

**Affiliations:** School of Pharmacy, Memorial University, St. John’s, NL A1B 3V6, Canada; saryal@mun.ca (S.A.); tcskinner@mun.ca (T.S.); bpbridges@mun.ca (B.B.)

**Keywords:** antioxidants, bioavailability, flavonoids, microbiome, natural products, neurodegeneration, neuroinflammation, nutraceuticals

## Abstract

Parkinson’s disease (PD) is a progressive neurodegenerative disorder that is characterized by a loss of dopaminergic neurons, leading to bradykinesia, rigidity, tremor at rest, and postural instability, as well as non-motor symptoms such as olfactory impairment, pain, autonomic dysfunction, impaired sleep, fatigue, and behavioral changes. The pathogenesis of PD is believed to involve oxidative stress, disruption to mitochondria, alterations to the protein α-synuclein, and neuroinflammatory processes. There is currently no cure for the disease. Polyphenols are secondary metabolites of plants, which have shown benefit in several experimental models of PD. Intake of polyphenols through diet is also associated with lower PD risk in humans. In this review, we provide an overview of the pathology of PD and the data supporting the potential neuroprotective capacity of increased polyphenols in the diet. Evidence suggests that the intake of dietary polyphenols may inhibit neurodegeneration and the progression of PD. Polyphenols appear to have a positive effect on the gut microbiome, which may decrease inflammation that contributes to the disease. Therefore, a diet rich in polyphenols may decrease the symptoms and increase quality of life in PD patients.

## 1. Parkinson’s

More than two centuries ago, in 1817, James Parkinson first described the disease with his namesake in his essay, An Essay on the Shaking Palsy [1]. Parkinson’s disease (PD) is a slow progressive neurodegenerative disorder that predominately affects dopaminergic neurons in the substantia nigra pars compacta (SNpc). PD is a debilitating condition caused by a mixture of genetic and environmental factors affecting various neuroanatomical sites and begins years before the diagnosis can be made [2]. The early disease can be divided into three stages: (i) Preclinical PD, the beginning of neurodegenerative processes but lack evident signs or symptoms; (ii) prodromal PD, the presence of signs and symptoms, however insufficient to define disease; and (iii) clinical PD, with diagnosis of PD based on the presence of classical motor signs [3]. Bradykinesia, rigidity, tremor at rest, and postural instability are the cardinal signs of PD [4]. However, non-motor symptoms such as olfactory impairment, pain, autonomic dysfunction (orthostatic hypotension, GI dysfunctions), impaired sleep, fatigue, and behavioral changes (depression, anxiety, apathy) are also observed [5].

### 1.1. Prevalence and Incidence of Parkinson’s Disease

The global burden of diseases study 2017 reported that the prevalence of PD globally was 8.52 (95% uncertainty interval (UI) 7.03–10.18) million and incidence was 1.02 (95% UI 0.85–1.22) million [6]. In 2017, 0.34 (95% UI 0.32–0.35) million people died from PD [7]. The male-to-female ratios of age-standardized prevalence rates of PD were 1.40 (95% UI 1.36–1.43) in 2016 and 1.37 (95% UI 1.34–1.40) in 1990 [8]. A combination of five separate cohort studies to estimate the prevalence of PD in North America shows overall prevalence to be 572 per 100,000 (95% confidence interval 537–614) among those aged ≥45 years and that that number is projected to increase to approximately 1,238,000 in 2030 [9]. In Korea, the annual incidence of PD was between 22.4–27.8 cases per 100,000 individuals, and the female-to-male ratio in the prevalence of PD was 1.6:1 while the incidence of PD was 1.4:1 [10]. Also, other PD prevalence studies show an increase in PD steadily with age and male gender [11,12,13].

### 1.2. Symptoms and Diagnostic Criteria

The movement disorder society (MDS) diagnostic criteria define supportive criteria, absolute exclusion criteria, and red flags. MDS uses a two-step approach, at first parkinsonism is defined, then checking whether parkinsonism is attributable to PD. In parkinsonism, bradykinesia must occur in combination with rest tremor, rigidity, or both. Parkinsonism caused by PD has a decline in either speed or amplitude as movements continue. A clinically established PD diagnosis requires the absence of absolute exclusion criteria, at least two supportive criteria, and no red flags. Absolute exclusion criteria include unequivocal cerebellar abnormalities (such as gait, limb ataxia, or cerebellar oculomotor abnormalities), downward vertical supranuclear gaze palsy, variable frontotemporal dementia, unequivocal cortical sensory loss, apraxia, or aphasia. It further includes parkinsonian features restricted to lower limbs for more than three years, anti-dopamine therapy, and absence of high-dose levodopa response. Supportive criteria are both motor and non-motor aspects. There should be a clear beneficial response to dopaminergic therapy, presence of levodopa-induced dyskinesia, rest tremor of a limb, and positive result on olfactory loss or cardiac sympathetic denervation. Red flags comprise early gait impairment, absence of progression of motor symptoms, early bulbar dysfunction, inspiratory dysfunction, severe autonomic failure in the first five years, unexplained pyramidal tract signs, and bilateral symmetric parkinsonism. Furthermore, included in these are recurrent falls due to impaired balance, disproportionate anterocollis, and absence of any common nonmotor features: Sleep dysfunction, autonomic dysfunction, hyposmia, or psychiatric dysfunction. This criterion does not include postural instability [14].

### 1.3. Pathology

The brain in PD is often unremarkable macroscopically with mild atrophy of frontal cortex and ventricular dilation. The characteristic morphological change in the PD brain appears in the transverse section of the brainstem with loss of darkly pigmented areas in the SNpc and locus coeruleus (see Figure 1 for diagram of the brain). This loss is correlated to the death of dopamine (DA) neuromelanin containing neurons in SNpc and noradrenergic neurons in locus coeruleus. The ventrolateral tier of neurons of pars compacta (A9) is mostly marked in neuronal loss in the substantia nigra (SN), while dorsal and medial neuronal cells are less susceptible [15].

Major pathological hallmarks of PD include the degeneration and death of melanin-containing neurons of SN and Lewy pathology. Lewy pathology comprises the formation of intracytoplasmic Lewy bodies (LB) with inclusions containing mainly α-synuclein and ubiquitin and Lewy neurites (LN), which are the neuronal projections of similar inclusions [16].

Braak and colleagues in 2003 staged PD pathology based on a semiquantitative assessment of LB distribution at postmortem, which revealed that LB pathology spreads rostrocaudally throughout the brain in a chronologically predictable sequence [17]. At Braak stages 1 and 2, LB lesions are seen in the dorsal motor nucleus (IX/X), reticular formation, and anterior olfactory nucleus where patients are considered asymptomatic/presymptomatic. Few non-motor features such as autonomic dysfunctions (constipation), olfactory dysfunction, and sleep-related dysfunction occur at these stages. The disease progression to Stage 3 involves SNpc with LB pathology, and neuronal loss is seen at melanized neurons. The pathology further extends to locus coeruleus and amygdala, ultimately reaching the temporal limbic cortex at Stage 4. During these two stages, typical clinical motor features start to appear. There is involvement of the entire neocortex and areas, including the prefrontal cortex and primary sensory and motor areas in stages 5 and 6 [17,18]. This system is criticized for being based only on the distribution of Lewy pathology but not on neuronal loss [19].

After the motor symptoms appear, nigral DA neuron loss increases to 60% or higher and strongly correlates with the severity of motor features and disease duration. This remarkable cell loss is the denervation of the nigrostriatal pathway, leading to diminished dopamine levels in the striatum. The reduction of dopaminergic signaling is considered responsible for the appearance of the cardinal motor symptoms in PD. Apart from the SNpc, widespread cell loss can be found in several subcortical nuclei, including the locus coeruleus, the nucleus basalis of Meynert, the dorsal motor nucleus of the vagus nerve, the pedunculopontine nucleus, the raphe nuclei, the hypothalamus, and the olfactory bulb [20]. Multiple non-dopaminergic neurotransmitter systems are affected, such as the cholinergic, glutamatergic, GABAergic, noradrenergic, serotonergic, and histaminergic systems [21].

### 1.4. Pathogenesis

In the onset of PD, multiple other processes are thought to be involved. The nuclear TUNEL labeling and chromatin condensation in PD brains suggest SN neurons may suffer a programmed cell death (i.e., apoptosis). Further examination of PD brains identified mitochondrial function changes, increased oxidative stress, lysosomal dysfunction, protein aggregation, impaired degradation, deposition of iron, inflammation, and glial activation [22] (see Figure 2).

#### 1.4.1. Reactive Oxidative Stress

Of the total oxygen supply to the body, the brain consumes about 20%. A significant portion of this oxygen converts into reactive oxygen species (ROS) from several sources in neurons and glia. At the mitochondrial level, the electron transport chain is the major contributor to ROS, while other sources include monoamine oxidase, NADPH oxidase, other flavoenzymes, and nitric oxide (NO) [23]. The oxidative damage in mitochondrial DNA results in impairment of the respiratory chain resulting in inhibition of complex I [24]. Also, ROS-mediated DNA damage, protein oxidation, and elevated levels of malondialdehyde, thiobarbituric acid, and 4-hydroxynonenal (HNE) are reported in SN and striatum of PD patients [25]. An increased alteration in transcriptional pathways, including nuclear factor erythroid 2-related factor 2, nuclear factor kappa B (NFkB), mitogen-activated protein kinase, glycogen synthase kinase 3β, and the decreased activity of glutathione, superoxide dismutase, and catalase with aging also correlates to the incidence of PD [26]. Oxidative damage is known to impair ubiquitination and degradation of proteins by the proteasome [27].

Likewise, reactive nitrogen species (RNS) such as NO, peroxynitrite (ONOO^−^), and nitroxyl radicals cause damage to the DNA, protein oxidation, and lipid peroxidation, affecting various cellular components, ultimately impairing cellular function. Such protein oxidation forms carbonyl, nitrotyrosine, and thiols of protein, negatively affecting its function [28]. Nitric oxide synthase (NOS) has three different isoforms, which generate NO:Neuronal NOS, localized in neurons, neutrophils, astrocytes.Endothelial NOS, present in vascular endothelium and astrocytes.Inducible NOS, expressed in macrophages, hepatocytes, glial cells, and neuronal cultures.

NO damages DNA via three ways: Direct reaction of RNS with DNA; inhibiting repair processes; and genotoxic species production such as H_2_O_2_ and alkylating agents [29]. Amines are nitrosated to nitrosamines by Dinitrogen trioxide (N_2_O_3_). N_2_O_3_ gets metabolized to various mutagenic alkylating species, thus damaging DNA and inhibits repairment [29,30].

##### Dopamine

The biosynthesis of DA occurs in the presynaptic terminals of DA neurons. Tyrosine enters the neuron through low- and high-affinity amino acid transporters and gets converted to dihydroxyphenylalanine (l-DOPA) in the presence of tyrosine hydroxylase, the rate-limiting enzyme. l-DOPA is further decarboxylated to DA, which is sequestered into vesicles via vesicle monoamine transporter 2, where a low pH environment prevents its autoxidation [31].

The enzymatic metabolism of DA is carried out by catechol-*O*-methyl transferase (COMT) and two isoforms of monoamine oxidase (MAO), MAO-A, and MAO-B. COMT is expressed predominantly by glial cells, which convert DA to 3-methoxytyramine (3-MT). Then, MAO reduces 3-MT to Homovanillic Acid (HVA) and is eliminated in the urine. On the other hand, MAO-A is mainly present in catecholaminergic neurons in SN, and MAO-B is found in astrocytes. MAO breaks down DA to 3,4-dihydroxyphenylacetaldehyde (DOPAL), which is later degraded by aldehyde dehydrogenase into 3,4-dihydroxyphenylacetic acid (DOPAC) and H_2_O_2_ [26]. Furthermore, dopamine’s catechol ring undergoes autoxidation to produce O_2_^•−^ and H_2_O_2_, which may react with metals like iron, producing toxic hydroxyl radical ^•^OH, through the Fenton reaction. Also, O_2_^•−^ may react with ^•^NO to form the toxic ONOO^−^. The electron-deficient dopamine quinones or semiquinones are also produced via autoxidation of dopamine, facilitated by metal ions [27]. Dopamine quinones contain a partial positive charge localized at C-5 atom of the catechol ring that can react with cellular nucleophiles such as sulfhydryl groups on both GSH and protein cysteinyl residues, covalently modifying the protein structures, which often result in protein inactivation and compromise cell survival [27,32]. Similarly, the dysfunction in the projections of dopaminergic neurons of the nigrostriatal DA pathway from the SN to the dorsal striatum would slowly lead to PD. Oxidative stress and oxidized DA results in the degeneration of the nigrostriatal pathway in PD [26].

Tyrosine and tyrosyl radical (TyrO^•^) interact with ROS and RNS through radical mechanisms and chain propagation reactions. One-electron oxidation of Tyr by heme ferryl ions forms TyrO^•^, which can cause apoptotic neuronal cell death by lipid peroxidation, crosslinking of protein(s) tyrosyl residues insitu, or coupling to produce o,o’-dityrosine (DT) and formation of peroxide adduct of tyrosine (TryOOH) or protein oxidation/nitration. Nitration is pathological as well as playing a role in signal transduction, as nitration of tyrosine residue through TyrO^•^ formation modulates phosphorylation (tyrosine kinase activity) and tyrosine hydroxylation (tyrosine hydroxylase inactivation). Consequently, it leads towards failure of dopamine synthesis and increased target protein degradation. Similarly, nitration serves as a blocker for radical-radical reaction, where TyrO^•^ scavenges NO^•^, NO_2_^•^, and CO_3_^•−^ [33].

##### Iron

Iron is an essential component in neurotransmitter synthesis, where the metal is a co-factor of tyrosine hydroxylase, which converts tyrosine to DA and further to norepinephrine. The chemical compound, called catechol, may potentially bind iron [34]. An excess of iron may lead to a vast increase in the production of free radicals, which overwhelms the natural defensive mechanisms and causes damage at several cellular levels. The SN of PD patients had elevated iron levels, in both pars compacta and pars reticulata [35].

Iron is transferred intracellularly via transferrin and stored bound to ferritin. Ferritin is comprised of heavy and light subunits whose ratio varies among tissues and in different cell types within the brain. Such variations affect iron interaction with other components in the cell, making some more vulnerable to oxidative stress. The heavy subunit possesses ferroxidase activity, converting Fe^2+^ to Fe^3+^, while the light subunit stabilizes the complex to remain in the storage form. Conditions such as a decrease in ferritin expression and degeneration of nigral melatonin neurons result in the increment of reactive Fe^2+^ iron pool. The reduced Fe^2+^ readily reacts with H_2_O_2_ to form OH^−^ radicals. Also, age-related iron increase and a leaky blood–brain barrier (BBB) leads to additional iron accumulation. In a healthy SN, the ratio of reduced to oxidized iron is 1:1; however, in PD patients, the ratio is increased to 1:3. Such an increment is toxic and is not found in other areas of the brain. The levels of free radicals intensify with decreased glutathione and autoxidation of DA stimulating a wide array of cytotoxic reactions such as protein misfolding, lipid peroxidation, glial cell activation, mitochondrial dysfunction, and α-synuclein aggregation [36,37]. A study using murine bone marrow-derived macrophages showed that dopamine affected cellular iron homeostasis by increasing the uptake of non-transferrin bound iron. It resulted in intracellular oxidative stress responses and increased the transcriptional expression of stress response genes such as heme oxygenase-1 and the iron export protein ferroportin [34]. Also, the iron released from Neuromelanin (NM) increases oxidative stress in mitochondria, disrupting its function and reducing ATP-dependent proteasomal activity of 26S (the proteasome involved in the ubiquitin system) [36].

##### Neuromelanin

NM is a black, brown insoluble substance formed from an oxidative metabolite of dopamine and norepinephrine. It reacts with lipids, pesticides, and toxins like paraquat, and metal ions like iron [38]. NM possesses a dual nature; inside neurons it protects cells from toxic effects of active redox metals, toxins, and excess of cytosolic catecholamines, while NM released by dying neurons activates neuroglia triggering neuroinflammation [39]. In a recent study, protein conjugated DA derived melanin oxidized GSH to GSSG with the production of H_2_O_2_, suggesting a potent prooxidant activity [40]. NM binds to iron and can be protected or damaging based on bound iron [37]. Under normal physiological conditions, excess iron can be sequestrated in ferritin and NM, and in this respect, NM has been described as neuroprotective [27]. When iron is in excess, NM may begin catalyzing free radicals [38]. Another potential toxic process involving NM is based on the high expression of major histocompatibility complex class I (MHC-I) in NM-containing organelles, which suggests the presentation of antigenic peptides by MHC-I on the neuronal membrane of catecholaminergic neurons containing NM [37]. It thus appears that iron, DA, and NM interact by multiple pathways [37].

##### Lipids

Lipid peroxidation leads to the formation of HNE. The level of HNE in dopaminergic cells are increased in SN shown by immunocytochemistry and in cerebrospinal fluid in PD [24]. Also, decreased polyunsaturated fatty acids and increased levels of malondialdehyde suggest increased lipid peroxidation levels in SN [27]. HNE is a highly reactive lipophilic α-beta alkenal that forms stable adducts with nucleophiles present in proteins like thiols and amines. HNE can also stimulate apoptosis by activation of caspases-8, -9, -3 and DNA fragmentation. Furthermore, it inhibits the NF-kB signaling pathway, reduces GSH level due to rapid consumption via GSH peroxidase, inhibits complex I and II of the mitochondrial respiratory chain, and stimulates PARP cleavage [24].

##### Glutathione

Glutathione is a thiol tripeptide made up of glutamate, cysteine, and glycine, which serves as an essential non-enzymatic antioxidant. Glutathione synthesis occurs in two steps. At first, γ-glutamylcysteine is formed from glutamate and cysteine in the presence of a γ-glutamylcysteine ligase, rate-limiting enzyme. Then, in the presence of glutathione synthase, glycine is added to γ-glutamylcysteine. Glutathione reductase reduces oxidized glutathione to maintain GSH levels. Nevertheless, oxidative stress alters the level of GSH. Glutathione conjugates with oxidized products of DA, l-DOPA, and DOPAC resulting in decreases glutathione levels. Such a decrease in SN may reduce the clearance of H_2_O_2_ formed by autoxidation of DA, ultimately promoting iron-induced OH^−^ formation [41]. The impaired complex I also leads to reduced GSH levels via increased ROS production. Such reduced GSH levels can occur due to decreased synthesis as a result of glutathione reductase inhibition or increment in the levels of glutathione disulfide (GSSG) and an altered ratio of GSH:GSSG [42]. In contrast, glutathione depletion in SN leads to the inhibition of mitochondrial complex I via thiol oxidation, ultimately reducing mitochondrial function [41].

#### 1.4.2. Mitochondria Dysfunction

Mitochondria play a significant role in energy metabolism and oxidative phosphorylation. The oxidative phosphorylation system is comprised of 5 major multi-subunit complexes: Complex I (NADH dehydrogenase-ubiquinone oxidoreductase), complex II (succinate dehydrogenase-ubiquinone oxidoreductase), complex III (ubiquinone-cytochrome c oxidoreductase), complex IV (cytochrome c oxidase), and complex V (ATP synthase) [43]. In the inner mitochondrial membrane, the transfer of electrons occurs through a series of protein complexes, and some protons from the matrix to intermembrane space are translocated, forming a proton gradient. Then, following the gradient protons flow back to the matrix simultaneously producing energy for ATP synthase to phosphorylate ADP to ATP. This process leads to the formation of ROS as a major byproduct. Also, premature election leakage in ETC complex I and Complex III to oxygen forms superoxide anions (O_2_**^−^**). Any dysfunction in ETC causes excessive ROS production, which is detrimental to the cells [44].

The first demonstration that mitochondria play a role in PD pathogenesis was after an individual consumed drug contaminated with the toxin 1-methyl-4-phenyl-1,2,3,6-tetrahydropyridine (MPTP). Studies show MPTP undergoes oxidation by MAO-B, forming 1-methyl-4-phenylpyridinium (MPP^+^), which enters DA neurons in SN via DA reuptake system. Subsequently, MPP^+^ inhibits mitochondrial ETC Complex I enzyme and NADH ubiquinone oxidoreductase leading to electron leakage and ROS production. Likewise, rotenone, pyridaben, trichloroethylene, and fenpyroximate are other inhibitors of complex I that induce PD. Rotenone being hydrophobic easily crosses biological membranes causing systemic inhibition of mitochondrial ETC [45].

Furthermore, PD-related genes, *PINK1*, *PARK2* (*Parkin*), *DJ-1*, and *LRRKS* encode proteins that regulate mitochondria and ROS homeostasis. *PINK1* is rapidly degraded in healthy mitochondria while in those exhibiting misfolded proteins, high oxidative stress, or reduced membrane potential, *PINK1* degradation is impeded, leading to *PINK1* accumulation in the outer mitochondrial membrane. *PINK1* recruits *Parkin* and induces E3 ubiquitin ligase activity. *Parkin* modifies mitochondrial membrane protein by adding ubiquitin chains, thus signaling for autophagy. Such autophagy, called mitophagy, results in mitochondria engulfment and degradation [46]. During oxidative stress, *DJ-1* translocates to the outer membrane and prevents MPP^+^ induced cell death, although the mechanism is unclear. The homozygous mutation of *DJ-1* increased mitochondrial oxidative stress, accumulation of α-synuclein, and DA oxidation [47].

Similarly, mitochondrial DNA (mtDNA) is an easy target of oxidation as histone proteins unprotect it. The proteins coded by mtDNA are involved in ETC. Mutation or deletions in mtDNA disturbs ETC and increases ROS formation. Also, nitrosative stress caused by mitochondrial toxins or mutated α-synuclein causes sulfonation on myocyte specific enhancer factor 2C inhibiting its transcriptional activity and expression of target genes such as peroxisome proliferator-activated receptor-gamma coactivator 1 alpha (*PCG-1α*). *PGC-1α* is a regulator of mitochondria biogenesis, whose failure leads to mitochondria dysfunction [44,48].

#### 1.4.3. α-Synuclein

α-Synuclein, a presynaptic neuronal protein, is a 140 amino acid (AA) long protein expressed at high levels. The N terminal (1–60) of seven highly conserved 11 AA repeat sequences form amphipathic α-helix allowing it to bind with membranes. The hydrophobic nonamyloid component (61–95) is amyloidogenic and responsible for protein aggregation. The C terminal (96–140) is polar, consisting of charged amino acid residues responsible for post-translational modification and mediates the interaction of α-synuclein with other proteins, ligands, and metal ions. α-synuclein exists in equilibrium between its soluble form, natively unstructured, and membrane-bounded form, α-helical structure [49]. The unique structure of α-synuclein can readily interact with anionic lipids, causing conformational changes and favor aggregation. In turn, these aggregate-prone soluble forms of α-synuclein can interfere with lysosomal and mitochondrial function, autophagy, vesicular homeostasis, and microtubule transport. Disruption of tubulins, kinesin- and dynein-containing complexes will interfere with anterograde and retrograde transport, respectively [50].

The widespread proteinaceous cytoplasmic inclusions, LB, and LNs contain filamentous aggregates of phosphorylated and ubiquitinated α-synuclein, causing misfolding and aggregation to occur. Formation of a non-fibrillar off-pathway and soluble transient pre-fibrillar intermediate called oligomers occurs. Oligomers convert into insoluble fibrillar aggregates with distinct cross beta-sheet conformation. Overexpression of α-synuclein results in cytotoxicity, as represented by several animal and cellular models. Aggregation pathway consists of: (1) Lag phase, rate limiting, leads to the formation of aggregation competent nucleus; (2) elongation phase, where the nucleus converts into protofibrils, and higher-order aggregating species; (3) in the stationary phase, the majority soluble protein converts into amyloid fibrils, and dynamic equilibrium exists between fibrils and monomers [49]. The rate of α-synuclein synthesis and clearance maintains the level of protein in the CNS, which occurs by direct proteolysis, chaperones, autophagy, and proteasome-mediated degradation. Failure in any of these paths leads to the accumulation of α-synuclein [50].

The collapse of the lysosomal degradation pathway to remove abnormal proteins promotes inclusion formation in cells. Individuals with a heterozygous mutation in *GBA1* have ~7% chance to develop sporadic PD. *GBA1* mutation leads to a reduction in glucocerebrosidase that can ultimately promote the accumulation of insoluble α-synuclein aggregates [51]. α-synuclein undergoes extensive post-translational modifications, including phosphorylation, nitration, and DA modification. In healthy brains, only a small fraction, approximately 4%, of total α-synuclein is phosphorylated at residue Serine-129 (Ser-129), while approximately 90% of phosphorylation at ser-129 is detected in PD brains containing LB. Such phosphorylation at ser-129 promotes oligomeric α-synuclein aggregation. α-synuclein phosphorylation is regulated by protein kinases, such as casein kinase, polo-like kinase 2, and dephosphorylation by phosphoprotein phosphatase, such as phosphoprotein phosphatase 2A, respectively [51,52].

Altered *SNCA* genes due either point to mutations (*p.A53T*; *p.A30P* and *p.e46K*) or whole locus multiplication lead to autosomal PD. Also, missense *SNCA* mutation leads to LB pathology, and *SNCA* duplication or triplication aggravates the accumulation of misfolded α-synuclein [53]. The overexpression of α-synuclein from the mutant form in primary dopaminergic neurons results in neuronal death. Also, PD toxins enhance the susceptibility of overexpression of α-synuclein mutant towards apoptosis mediated cell death. Twenty-two different genes with causal mutations are associated with PD or risk factors. Mutations in *A53T*, *E46K*, and *H50Q* accelerates α-synuclein fibrillation while *A30p*, *A53E*, and *G51D* delay the aggregation. *A53V* also accelerates α-synuclein aggregation and promotes early oligomerization [51]. α-synuclein is associated with SNARE complex and acts as a chaperone regulating degradation, distribution, and maintenance of SNARE, directly involved in dopamine release [54]. Oligomers and amyloid of α-synuclein are neurotoxic. Exogenously added α-synuclein could internalize and seed endogenous monomeric α-synuclein into LB, transmitting from one cell to another, thus spreading PD [50].

#### 1.4.4. Neuroinflammation

Neuronal cell death occurs via two mechanisms: Cell-autonomous, where cell death occurs by the accumulation of intrinsic damage of degrading neurons, and non-cell-autonomous, where neuronal degeneration occurs via pathological interaction with glial cells (microglia, astrocytes) or infiltration of peripheral immune cells (macrophages, lymphocytes) [55]. Microglia, innate immune cells in the brain, in the presence of pathogens or tissue damage, induce complex immune responses via increased expression of toll-like receptors, pro-inflammatory mediators, and activation of peripheral immune cells, and initiates oxidative stress to restore tissue homeostasis [56].

Activated microglia undergo morphological changes and phenotypical alteration in gene expression and activation of signaling molecules. These activated microglia have dichotomous roles in neuroinflammation in that they can both mediate the inflammatory response by producing mediators that function to clear the source of the inflammatory stimuli, and they can regulate the inflammatory response by perpetuating the pro-inflammatory response through the continued release of inflammatory products, which function to activate and regulate microglia and astrocyte responses. Conversely, when the inflammatory stimuli have been removed, microglia can function to promote neurogenesis through the release of neurotrophins and anti-inflammatory cytokines, potentially leading to neuroregeneration and wound healing within the SN and striatum [57].

Cytokines, chemokines, and other inflammatory mediators trigger microglial activation and contribute to nigrostriatal pathway injury. Elevated levels of inflammatory cytokines, TNF-α, IL1beta, IL-2, IL-4, IL-6, interferon γ (IFNγ), nitric oxide synthase (NOS), and ROS are evident in postmortem examination of SN. In neurons, activation of NFκB promotes their survival, while in glial cells, NFκB mediates proliferation and activation. NFκB promotes apoptosis and expression of *TGFβ1* and cyclopentenone prostaglandins and transcription of genes such as *Bax* and *p53* [57,58,59]. Microglia are the main MHC class II-expressing antigen-presenting cells in brain parenchyma with neuronal damage [58]. T cell infiltration is also increased in postmortem brain tissue of PD patients [60]. A study shows that CD4^+^ T cells from PD patients react specifically with α-synuclein derived MHC class II epitopes [61]. Also, another study of MPTP induced neurotoxic mice lacking T cells (CD4^+^ T cells) shows a decline in dopaminergic cell death and microglial activation suggesting that CD4^+^ T cells affect PD disease pathology, potentially through MHC class II-expressing parenchymal microglia [60].

Microglial activation promotes further α-synuclein pathology by increased NO production, which can induce the nitration of α-synuclein in neighboring neurons and result in cell death [62]. Conversely, α-synuclein pathology, and dysregulation of monocytes can also induce excessive inflammatory responses to α-synuclein [63]. NM also triggers microgliosis, microglial chemotaxis, and microglial activation. NM activates NFkB by phosphorylation and degradation of inhibitor protein B and leads to the upregulation of TNF-α and NO [64].

### 1.5. Risk Factors

#### 1.5.1. Genetic Risk Factors

The mendelian genetic transmission includes autosomal dominant transmission and recessive transmission. In autosomal transmission, the *SNCA* gene with six-point mutation (*53T*, *E46P*, *A30P*, *H50Q*, *G51D*, and *A53E*) is linked to increased propensity of α-synuclein to form oligomers or fibrils. Similarly, *LRRKS* implicates vesicle trafficking, autophagy, and mitochondrial function. A mutation in the *VPS35* gene is associated with retrograde vesicle transport and cathepsin D trafficking leading to α-synuclein degradation. Also, the *CHCHD2* gene is linked to the mitochondrial respiratory complex [65]. In recessive transmission, *Parkin* interacts with *PTEN*-induced putative kinase 1 (*PINK1*) and *FBXO7* for the degradation of malfunctioning mitochondria. Also, *PINK1* interacts with *Parkin* for recruiting to mitochondria. These genes are involved in mitophagy pathways and overexpressing wild-type *PINK1* rescues stress-induced apoptotic death. *DJ1*, a chaperone protein, regulates the *PINK1*-*Parkin* translocation to mitochondria and prevents the aggregation of α-synuclein. Upon mutation, it alters the shape of mitochondria, and ROS levels are increased. *ATP13A2* is responsible for a juvenile-onset syndrome characterized by parkinsonism, dystonia, and supranuclear palsy. *PLA2G6* is associated with neurodegeneration and brain accumulation of iron. *FBXO7* directly interacts with *Parkin* and *PINK1* for mitophagy and when mutated, it is mislocalized to the cytosol. *NAJC6* coding for auxilin is involved in clathrin uncoating and synaptic vesicle formation and recycling [65]. Glucocerebrosidase (*GCase*) gene (*GBA*) mutations are responsible for Gaucher disease, a lysosomal storage disorder. Heterozygous *GBA* mutation carriers also have an increased risk of developing PD. *GBA* mutations are *N370S*, *L444P*, and *E326K*. They lead to dysfunction of the autophagy-lysosomal pathway, followed by a decreased degradation of α-synuclein. *GBA1* mutations perturb healthy mitochondria functioning by increasing generation of ROS and decreasing adenosine triphosphate (ATP) production, oxygen consumption, and membrane potential. *GBA1* mutations also lead to the accumulation of dysfunctional and fragmented mitochondria [66]. Non-mendelian genetic transmission includes SNP of the *SNCA*, *MAPT*, and *LRRK2* genes, but additional genes, such as *HLA-DRB5*, *BST1*, *GAK*, *ACMSD*, *STK39*, *MCCC1/LAMP3*, *SYT11*, and *CCDC62/HIP1R* were also reported [67].

#### 1.5.2. Environmental Risk Factors

Neurotoxins in the brain lead to oxidative stress and disruption of neurotransmission that results in detrimental effects in the basal ganglia. Several chemical elements, such as iron and copper, contribute to oxidative stress. Copper acts via the Fenton-Haber-Weiss reaction and 6-OHDA redox cycle [68], causing the reduction of dopamine, α-synuclein aggregation, and the reduction of superoxide dismutase 1 [69]. Manganese (Mn) exposure can change oxidative parameters such as decreased glutathione levels, enhanced lipid peroxidation, and protein oxidation [70]. Lead (Pb) enters the brain by mimicking calcium via calcium channels and leads to swelling and loss of neurons in the CNS and peripheral nervous system. Pb causes a decline in voluntary muscle movements [71]. Also, Pb may exacerbate PD-related neural dysfunction, resulting in impaired cognition [72]. Mercury (Hg) can reduce the number of neurons present in the brain resulting in tremors and loss of voluntary muscle movement [71].

Pesticides can also target the SN. Dieldrin, an organochlorine pesticide, causes neurotoxic damage to the dopaminergic system [73]. Rotenone, an organophosphate, increases α-synuclein aggregation and inhibits mitochondrial complex I [74]. Various illicit substances can elevate ROS leading to neurotoxicity. Methamphetamine decreases the integrity of DA neuron terminals in basal ganglia and decreases DA and dopamine transporter expression. High doses of amphetamine may cause damage to dopaminergic neurons and axon terminals within the human brain. Cocaine binds to DA transporters causing short-term DA inhibition, and cocaine addiction can lead to iron dysregulation [75].

## 2. Polyphenols

Polyphenols are the secondary metabolites of plants characterized by at least two phenyl rings and one or more hydroxyl substituents [76]. On the basis of the nature of the carbon skeleton, polyphenols are divided into four major classes: Phenolic acids, Flavonoids, Stilbenes, and Lignans [77].

### 2.1. Phenolic Acids

Based on the derivatives of benzoic acid and cinnamic acid, phenolic acids are further broadly divided into two classes. The first class includes Hydroxycinnamic acids, which are present in all parts of fruit with maximum concentration in outer ripened parts. The free form of hydroxycinnamic acids constitutes *p*-coumaric, caffeine, ferulic, and sinapic acids. In contrast, the bound forms are glycosylated derivatives or esters of quinic acid, shikimic acid, and tartaric acid. In comparison, hydroxybenzoic acids are less common. They constitute a smaller portion of edible plants and generally form a component of a complex structure such as hydrolyzable tannins [78].

### 2.2. Flavonoids

Flavonoids alone make up 60% of dietary polyphenols with 4000 varieties [79]. The chemical structure comprises of flavonoid diphenyl-propane skeleton, made of two aromatic rings (A and B) bound together by three carbon atoms that form an oxygenated heterocycle (ring C) [78,80]. Based on the degree of unsaturation and the substitution pattern, different flavonoid classes are distinguished: Flavones, flavonols, flavanones, flavan-3-ols, anthocyanins, dihydroflavonols, and isoflavones, as well as the biogenetic intermediate chalconoid forms. In natural sources, they may occur in free forms (aglycones), as glycosylated or acylated derivatives, and as oligomeric and polymeric structures such as the flavan-3-ol-derived condensed tannins (or proanthocyanidins) [81]. The various classes of flavonoids differ in the level of oxidation and pattern of substitution of the pyrene ring, whereas individual compounds within the classes differ in the pattern of substitution of benzene rings. In flavonoids, the B-ring links to the C-ring at the C2 position (see Figure 3), while the B-ring of isoflavonoids is substituted at position C3. Biflavonoids comprises two identical or non-identical flavonoid units conjoined through an alkyl- or alkoxy-based linker [82].

### 2.3. Stilbenes

The chemical structure of stilbenes is made up of a 1,2-diphenylethylene nucleus with hydroxyl substitution on the aromatic rings. They are present only in lower quantities in the human diet, most common being *trans*-resveratrol and its natural glycoside, *trans*-piceid. The naturally occurring phytoalexin, resveratrol, is produced by a wide variety of plants as a response to stress, injury, ultraviolet (UV) irradiation, and fungal (e.g., *Botrytis cinerea*) infection [78,80,83].

### 2.4. Lignans

Lignans are derived from the oxidative dimerization of two or more phenylpropanoid units [84]. Based on dimer linkage, lignan can be traditionally classified into two types: Classical lignans, constituting a ß-ß_0_ dimer linkage, and neolignans with different couple patterns. Classical lignans comprise six subtypes: Dibenzylbutanes (CL1), dibenzylbutyrolactones (CL2), arylnaphthalenes (CL3), dibenzocyclo-octadienes (CL4), substituted tetrahydrofurans (CL5a-c), and 2,6-diarylfurofurans (CL6), and neolignans consist of 15 subtypes (NL1 to NL15) [85].

## 3. Mechanisms of Neuroprotection

There is currently limited data concerning the mechanisms of neuroprotection offered by a polyphenol diet, particularly in PD patients. Several studies have experimented with various models of PD and different chemicals inducing PD-like symptoms. One such chemical is 1-methyl-4-phenyl-1,2,3,6-tetrahydropyridine (MPTP). This lipophilic molecule crosses the blood–brain barrier and is converted by monoamine oxidase B to 1-methyl-4-phenylpyridinium (MPP^+^) via biotransformation [86]. This conversion generates reactive oxygen species (ROS), causing harmful oxidative stress [87]. MPP^+^ is concentrated by dopaminergic neurons in the substantia nigra pars compacta. Following its uptake and concentration, MPP^+^ becomes concentrated in the mitochondria at levels that inhibit Complex 1 of the mitochondrial respiratory chain. Complex 1 deficit is also seen in PD patients. MPTP causes an increase in α-synuclein, activation of astrocytes, and destruction of the blood–brain barrier. These changes, in turn, result in inflammation, oxidative stress, and decreased dopaminergic neurons, similar to what is seen in PD [88]. 6-hydroxydopamine (6-OHDA) is another popular chemical used in PD studies. This molecule also induces the degeneration of dopaminergic neurons, as it is taken up by neurons and then accumulates in the cytosol. Similar to MPTP, 6-OHDA forms ROS that cause oxidative stress [89]. MPTP and 6-OHDA in particular have been useful in experimental studies of PD, as their effects are similar to the natural effects of PD seen in patients, despite their rapid onset of action. Knowing the mechanisms of action of PD-inducing chemicals allows researchers to study the mechanisms of action of possible therapeutic options in comparison.

The role of diet and polyphenols in PD has recently been reviewed by other research groups [90,91,92]. In terms of general diet, the Mediterranean Diet and Asian Diet have been shown to prevent a variety of pathologies associated with aging. In fact, it is the polyphenols in these diets that are associated with the greatest reduction in neurodegeneration. Dietary polyphenols (see Figure 4) have multiple targets, including the inflammatory pathway, ROS, and alterations in proteostasis in disease states associated with amyloid dysfunction [93]. This could likely make them valuable in possible prevention strategies. All natural polyphenols have radical scavenging effects due to their phenolic ring(s) and aromaticity, making them an interest for diseases involving oxidative stress toxicity. However, it seems as though a main mechanism for reducing levels of ROS is through activating enzymatic antioxidant defenses already present in the body, such as glutathione, catalase, and superoxide dismutase [94].

### 3.1. Phenolic Acids

Not many phenolic acids have been studied in the context of PD or general neurodegenerative disorders. A phenolic acid that has been studied is paeoniflorin, which is the main bioactive component of the herb Paeoniae Alba Radix [95]. Paeoniflorin is thought to have no direct effect on the dopamine receptor but to exert its effects by activating adenosine A1 receptors. A1 receptor activation improves dopaminergic deficits and receptor antagonism further worsens dopaminergic deficits in MPTP models. Thus, pretreatment with paeoniflorin improves PD-like symptoms and reduces the loss of tyrosine hydroxylase (TH), similarly to other classes of polyphenols [96,97]. Paeoniflorin has been hypothesized to have effects on the autophagy-lysosome pathway, which is responsible for removing misfolded proteins, therefore having potential for neurodegenerative conditions. After MPTP exposure, paeoniflorin restores cell viability and inhibits Ca^2+^ influx, reducing apoptosis [98]. Gallic acid, another phenolic acid found in chestnuts, cloves, and red wine, has protective effects on cell viability and reduces nuclear apoptosis in 6-OHDA-treated cells. The gallic acid ester metabolites (*n*-propyl gallate (PG), methyl gallate, *n*-octyl gallate, and *n*-dodecyl gallate) have shown the most promise for protective effects. It has been demonstrated that these metabolites show more statistically significant effects compared to other gallic acid derivatives. PG in particular has the most significant effects in almost all measured categories of neuroprotection. Gallic acid and its derivatives decrease toxicity by attenuating ROS increase, decreasing oxidized glutathione (GSSG), increasing reduced glutathione (GSH), and decreasing Ca^2+^ concentration. With these changes, only the ester metabolites have significant results [99].

### 3.2. Flavonoids

A larger intake of dietary flavonoids, anthocyanins in particular, is associated with lower PD risk in humans. However, total flavonoid intake (including anthocyanins, quercetin, epicatechin, and some proanthocyanidins) has only been associated with a lower risk of developing PD in men. The decrease in PD risk for women is not statistically significant [100]. Flavonoids are potentially the most studied class of polyphenols and have the most subclasses. The majority of flavonoids are found in berries, though berry leaves tend to have higher flavonoid and phenolic content [101].

Flavonoids have protective effects such as the prevention of dopamine reduction, ROS formation, TH protein loss, and mitochondrial Complex 1 deficits [102]. As well, flavonoids increase cell viability, prevent apoptosis, and decrease proinflammatory cytokines. Kaempferol, found in safflower along with its derivatives and a standardized safflower flavonoid extract, increases cell viability and decreases α-synuclein aggregates following neurotoxicity. It also prevents loss of TH and decreases GFAP, a pro-inflammatory cytokine, which indicates a decrease in astrocyte activation [103,104]. Similarly, quercetin increases cell viability and prevents apoptosis after MPTP-treatment, in addition to decreasing pro-inflammatory cytokines such as TNF-α and IL-6. Fisetin also reduces the production of these proinflammatory cytokines, along with reducing NO and PGE2, doing so by inhibiting inducible nitric oxide synthase (iNOS), COX-2, and IL-1Beta. Other factors involved with the inflammatory response include NFκB and p38 mitogen-activated protein kinase (MAPK), which are activated through IκB degradation and translocation of the NFκB p65 subunit. Fisetin and quercetin attenuate the inflammatory response through inhibition of this pathway, with fisetin specifically inhibiting the degradation of IκB [105,106,107]. Fisetin and quercetin maintain glutathione levels during periods of oxidative stress and are the only two flavonoids to demonstrate this property, indicating that specific polyphenols have differing mechanisms of neuroprotection [108].

Flavonoids are thought to have an indirect effect on the Keap1-Nrf2-ARE antioxidant response pathway. Nuclear E2-related factor 2 (Nrf2) functions to regulate the gene expression of molecules containing antioxidant response elements (ARE) and is controlled by the Kelch-like ECH associating (Keap1) protein, which acts as a sensor for redox balance in cells. Combined, the Keap1-Nrf2-ARE pathway maintains internal homeostasis under harmful conditions, often seen in many neurodegenerative disease states [109]. Under conditions of stress, Nrf2 is released from Keap1 and translocates to the cell nucleus where it forms a complex with a small musculoaponeurotic fibrosarcoma oncogene homolog (Maf) protein. This activates ARE-dependent gene expression of antioxidant proteins such as superoxide dismutase (SOD), heme oxygenase-1 (HO-1), and glutathione reductase (GR) [110]. Nrf2 is especially valuable in regulating redox homeostasis as it activates antioxidants, anti-inflammatory molecules, phase I and II metabolizing enzymes, efflux transporters, and free radical scavengers [111]. In models of neurodegeneration, many flavonoids such as hesperetin, farrerol, apigenin, and luteolin have displayed neuroprotective effects through the Keap1-Nrf2-ARE pathway [112,113,114].

Apigenin and luteolin (both flavones) prevent TH protein loss and improve antioxidant defense in MPTP-treated mice. The aforementioned polyphenols also significantly decrease levels of TNF-α compared to MPTP-treated mice, displaying potential for reducing the severity of the inflammatory cascade [115]. Certain flavonoids including fisetin, quercetin, luteolin, and isorhamnetin (a flavonol) are able to induce neurite outgrowth, though all are dose dependent and require a continuous presence of flavonoid. The flavonoids that induce neurite outgrowth do so through the Ras-extracellular signal-regulated kinase (ERK) signaling pathway [116]. In contrast, epigallocatechin gallate (EGCG), a catechin and flavan-3-ol found mostly in green tea, binds to α-synuclein fibril aggregates to restructure the fibrils and remodel beta-sheet-rich amyloid structures, reducing their toxicity. The restructured fibrils are smaller and more amorphous protein aggregates that confer a reduced level of toxicity, which has potential to be an asset in protein misfolding diseases, of which PD is included [117]. Polyphenols continue to be studied as they are observed to function as prevention against the PD-like deficits seen with experimental neurodegenerative models.

### 3.3. Stilbenoids

Polyphenols have been studied for their neuroprotective effects as they reduce oxidative stress, modulate signaling pathways, and inhibit the formation of amyloid aggregates [118]. These effects have been particularly studied with resveratrol, a stilbenoid found in grapes and blueberries. In rodent models of PD, administration of resveratrol, both intravenously and orally, protects against motor coordination impairment, reduces dopaminergic neuron degeneration in the substantia nigra, and decreases the expression of proinflammatory mediators COX-2 and TNF-α [119,120]. ROS and amyloid beta peptide, often seen in neurodegenerative diseases, can induce apoptosis in a variety of cells, including those in the substantia nigra. This toxic effect can be blocked by resveratrol. Resveratrol induces the MAP Kinase signaling pathway and early growth response 1, a transcription factor that could potentially regulate synaptic plasticity-related to learning and memory [121]. Resveratrol has been shown in PC12 cells to have higher levels of IκBα, an important inhibitor of the NFκB inflammatory pathway, and lower levels of nuclear p65, which helps regulate NFκB function. The exact molecular mechanisms for this effect are still unclear, however it does demonstrate the proposed anti-inflammatory effects through downregulation of NFκB [122]. Resveratrol is implicated in the prevention of cellular loss in the substantia nigra due to MPP^+^ toxicity. It also prevents striatal dopamine loss and TH loss, both of which are typically seen in the dopamine neurons of PD patients [87]. Outside of rodent research, resveratrol increases survival rate and longevity and improves MPTP-induced acetylcholine inhibition and decreases H_2_O_2_ and NO elevations in *D. melanogaster* flies [123]. Resveratrol has been suggested to show synergistic effects when given as treatment with flavonoids, as both polyphenols have differing cellular structures and thus could work in different cell compartments. This information could prove to be useful in dietary polyphenol studies, as polyphenols found in food tend to be from multiple classes [121].

### 3.4. Lignans

Lignans are commonly found in whole grains, seeds, nuts, and legumes. Schisandrin is a fairly popular lignan in traditional Asian and Eastern European medicine found in the fruit of the *Schisandra chinensis* plant. It includes subgroups schisandrin A, B, and C and has been observed in cellular models to exhibit protective effects against glutamate-induced neurotoxicity. In an experimental study with rat cortical cells, pretreatment with schisandrin had significant effects in glutamate-treated cells. Schisandrin reduced levels of ROS, restored mitochondrial membrane potentials, and reduced Ca^2+^ influx. Combined, these effects decreased cytochrome C release in schisandrin-treated cells, thus reducing cell apoptosis. Cellular defense mechanisms that had otherwise been reduced by glutamate, such as GSH and caspase, also avoided depletion when treated with schisandrin. Schisandrin A has a neuroprotective effect similar to other classes of polyphenols through its actions of inhibiting oxidation, inflammation, and autophagy. Schisandrin increases levels of TH expression and reduces inflammatory cytokines, however reduction of IL-1Beta and TNF-α were not found to be significant in this case. As in PD, MPTP can induce significant decreases in *LC3-II*, *beclin1*, *parkin*, and *PINK1*, all of which significantly resist depletion after treatment with schisandrin A, suggesting it enhances autophagy [124,125,126]. As another lignan, 7-hydroxymatairesinol, found in strawberries and sesame seeds, was recently shown to prevent striatal terminal loss and neuroinflammation in the substantia nigra, and decrease levels of microglial and astrocyte markers. Like other polyphenols, 7-hydroxymatairesinol reduces TNF-α and iNOS, as indicated by its ability to reduce the inflammatory process. A potential explanation for this is speculated to be the estrogen-like anti-inflammatory activity of 7-hydroxymatairesinol. Treatment with 7-hydroxymatairesinol in 6-OHDA-treated rats showed no changes in neuronal survival in dopaminergic cells, although it did notably reduce the inflammatory process in the region [127]. Sesamin, sesamol, and sesaminol, all found in sesame oil, have been studied for their potential neuroprotective effects in MPTP-treated mice. Small doses of sesamin protect cells from oxidative stress by reducing ROS production, decreasing iNOS and IL-6 expression in microglial cells, and increasing superoxide dismutase and TH levels. Pre-treatment with sesamin was also found to reduce the catalase increase from MPTP-treated cells, indicating its potential for direct antioxidant activities [128].

### 3.5. Notable Polyphenols

Curcumin, a popular polyphenol found in turmeric, has various effects that have been studied in relation to Alzheimer’s Disease but could also be of use in similar neurodegenerative diseases. Curcumin has antioxidant and anti-inflammatory properties as well as effects on neurogenesis and neuroprotection. It inhibits oligomerization of amyloid-beta protein and it affects the heat-shock protein system, which is important in removing misfolded proteins [129]. Curcumin, along with resveratrol, is called a hormetic phytochemical due to its ability to activate adaptive stress response pathways that increase cellular resistance to injury [130]. It is likely that keto-enol-enolate equilibrium of the heptadienone moiety of curcumin is what determines its beneficial properties, as it increases curcumin’s ability to scavenge ROS. Curcumin also decreases TNF-α production in the inflammatory pathway by inhibiting an acetyltransferase enzyme involved with the transcription process of TNF-α, as well as mediating the methylation of the TNF-α promote [131]. As with other polyphenols, curcumin is lipid soluble and thus can enter the blood brain barrier and inhibit iNOS, inhibiting microglial activation. Notably, curcumin inhibits apoptosis, COX-2, heat shock protein 60/heat shock factor-1 (HSP60/HSF-1), and the phosphoinositide 3-kinase/protein kinase B (PI3k/Akt) pathway, all of which play a role in microglia activation. Curcumin also induces the activation of antioxidant genes within the antioxidant response element (ARE) mechanism, such as heme-oxygenase-1, by their activators, such as nuclear factor erythroid 2-related factor 2 [132]. Curcumin, while not falling into any particular class, is still a notable polyphenol with potential in neurodegenerative disease.

## 4. Bioavailability

The bioavailability of dietary polyphenols is understood to be generally quite low. In particular, polyphenols with a higher molecular weight are usually too large to be absorbed in their original unmetabolized form. Therefore, the antioxidant effect of polyphenols has largely been thought of in relation to the digestive tract due to their higher concentration in the gut compared to the rest of the body [133]. Polyphenols are receiving attention for their possible effect on the gut microbiome, which in turn has been investigated for its potential effect in the brain. As polyphenols have variable chemical structures, they also have variable pharmacokinetic properties [77].

### 4.1. The Gut Microbiome and Brain Connection: Current Perspectives

There is much interest in the bidirectional microbiota-gut-brain axis and its potential implications in neurological disorders, as recently reviewed by Raval et al., 2020 [134]. It has been observed that a complex gut microbiome promotes the maintenance of microglia, while an absence results in deficits. As such, the gut microbiome has a role in the immune response of the brain and central nervous system (CNS), in which the enteric nervous system is thought to be a major key. It is suggested that a depletion of intestinal microbes results in a reduced immunological response from microglia and that sufficient microbe diversity is required for comprehensive microglial maturation [135,136]. Interestingly, the development of microbiota occurs parallel to neurodevelopment, alluding to their neurological role. Potential mechanisms by which microbiota are thought to possibly affect the CNS involve acting through altered microbial composition, immune activation, neural pathways, tryptophan metabolism, gut hormonal response, and/or bacterial metabolites [137]. Though not studied in humans, the presence of intestinal bacteria in mice is suggested to influence the development of anxiety-like behavior, motor control, memory, and learning [138]. An absence of microorganisms in mice resulted in increased motor activity, decreased anxiety-like behavior, and an increased turnover rate of noradrenaline, dopamine, and serotonin in the striatum. Increased BBB permeability is also the result of an absence or decrease in gut microbiota. BBB permeability can be partially determined by endothelial tight junctions, which are mostly decreased and disorganized in the absence of microorganisms. Notably, colonization with microbiota reversed these effects [139,140]. In healthy women, consumption of a fermented milk product containing probiotics (also called psychobiotics) enhanced cognition and reduced stress responses, though the mechanism of action was not addressed [141,142]. Probiotics have yet to become a reliable treatment option, however they do warrant further investigation as they have great implications for brain activity, neurodegenerative diseases, and their connection with the gut microbiome.

When discussing the elderly, who make up the majority of PD patients, it is important to remember that malnutrition is very common. Malnutrition is linked with poor food digestion and absorption, resulting in nutrition-deficit diets that negatively affect the gut microbiome. Elderly patients also tend to have chronic systemic inflammation, aptly called “inflammaging”, which could be a risk factor for the increased prevalence of degenerative diseases [143]. Intestinal microbiota changes associated with aging result in gut microbiota becoming weakened against environmental factors. Changes in the dominant bacterial species, a decline in beneficial microorganisms, an increase of facultative anaerobic bacteria, and a decrease in short chain fatty acids have been observed in the elderly [144]. Due to the many confounders with elderly patients, it is difficult to find a parallel link between age-related changes in gut microbiota and cognitive decline [145]. However, a decline in microbial complexity may possibly go hand-in-hand with the age-associated increased risk of neurodegeneration, thus making this a possible avenue of study for PD.

Gut inflammation has been demonstrated in PD. PD patients have significantly increased levels of TNF-α, IL-6, IL-1beta, and IFN-gamma (all proinflammatory cytokines) in the ascending colon. Enteric glial markers that are known to be upregulated by proinflammatory cytokines, such as GFAP and S100-beta, are also increased in PD patients [146]. In an experimental study with α-synuclein overexpressing mice, those with a more complex microbiota displayed decreased motor capabilities and a quicker progression of motor decline. An absence of microorganisms resulted in delayed motor decline, fewer α-synuclein aggregates in key brain areas (the caudoputamen and substantia nigra), and reduced GI defects [147]. There are also data that α-synuclein is overexpressed in the enteric nervous system of PD patients and may potentially migrate to the CNS via the vagal nerve or vice versa [136,148]. Gut inflammation in PD causes many symptoms, with gastrointestinal dysfunction affecting a majority of PD patients and idiopathic constipation being a risk factor for PD. When comparing fecal composition, PD patients have a decreased amount of *Prevotellaceae* bacteria. A decrease in *Prevotellaceae* is associated with decreased mucin synthesis and increased gut wall permeability, potentially leading to increased toxin exposure [149]. All in all, the information presented here confers that gut microbiota promote both motor and intestinal dysfunction similar to that of PD.

### 4.2. Polyphenols and the Gut Microbiome

When discussing polyphenols and gut microbiota, it is important to mention polyphenol metabolism. The chemical structure of polyphenols is what determines their ability to undergo biotransformation, which is displayed in the following examples. After the ingestion of flavonoids, their sugar moieties are cleaved from their phenolic backbone in the small intestine, where they are absorbed. In comparison, flavan-3-ols are never glycosylated but can be acetylated by gallic acid. Proanthocyanidins have a higher molecular weight so they are very unlikely to be absorbed in their native form in the small intestine. Hydroxycinnamic acids commonly have ester bonds linking them to compounds in which no human tissue esterases can break. Thus, they are majorly metabolized in the colon by colonic microbiota [150]. A small number of polyphenols are directly absorbed into the small intestine after deconjugation reactions. In the small intestine, polyphenols undergo extensive Phase I and Phase II reactions, allowing metabolites to be released into systemic circulation. In the large intestine, metabolism differs as polyphenols are enzymatically degraded by bacteria. Microbial-derived metabolites may be absorbed or excreted by the feces. If they are absorbed, they move through the portal vein to the liver where they may undergo Phase II biotransformation and absorb into systemic circulation or become eliminated in the urine. Microbial glucuronidase and sulphatase have the ability to deconjugate Phase II metabolites further [151,152]. In an experimental study, wine polyphenol-treated rats had a significantly higher abundance of *Lactobacillus* and a lower abundance of *Clostridium*. Although not statistically significant, *Bifidobacterium* presence increased and *Bacteroides* presence decreased after polyphenol treatment. *Lactobacilli* and *Bifidobacteria* are considered beneficial, while *Clostridia* is considered detrimental for the intestines [153]. The possibility of polyphenols affecting the gut microbiota and, as such, acting as probiotics has been discussed. Supplementation of EGCG and resveratrol in overweight individuals significantly decreased the amount of *Bacteroidetes* and *F. prausnitzii* compared to placebo in men but not women. It is noteworthy though that there was a greater prevalence of *Bacteroidetes* in men than women at baseline. For both groups, fat oxidation and skeletal muscle mitochondrial oxidative capacity were increased with polyphenol supplementation [154]. This could possibly be of relation to a previously mentioned study where a polyphenol-rich diet decreased PD risk in men but not women.

### 4.3. Polyphenols, Plasma, and the Brain

There has not been extensive research concerning the bioavailability of polyphenols in the brain. However, quite a bit of research has been done to examine general polyphenol bioavailability. Most polyphenols are excreted within 24 h of their intake. Caffeic acid, for example, has a much higher percentage of urine excretion than quercetin, meaning it is more likely to pass through the kidneys and digestive tract [77]. After the consumption of coffee, caffeic acid, along with its metabolites, and chlorogenic acid (all phenolic acids) were the main polyphenols recovered in urine [155]. Caffeic acid is also absorbed much more than chlorogenic acid from the small intestine, though both are thought to be metabolized extensively in the body and it is thought that only small amounts are found in the urine [156]. Ferulic acid (a phenolic acid) is also excreted considerably in urine, with a considerable amount being the glucuronide metabolite [157]. Ferulic acid, genistein (an isoflavonoid), and hesperetin (a flavanone) are taken up into the small intestine of rats at similar rates and the resulting secretion of metabolites by the small intestine and liver was found to depend on the rate of uptake. The higher the rate of absorption, the lower the rate of secretion [158]. Thus, a low rate of intestinal and biliary excretion confers a higher concentration present in the blood. Gallic acid (a phenolic acid) has not been extensively studied, although from the studies that have been conducted, it appears to be well absorbed. In a survey of 97 bioavailability studies, gallic acid was found to be the most efficiently absorbed in comparison to all other polyphenols [159].

The stilbene resveratrol and its metabolites have all been found to increase in concentration in the plasma and CSF over the course of treatment with resveratrol. Concentrations in plasma were expectedly higher than in the CSF, though all were notable for the polyphenol’s ability to enter the body [160]. On the other hand, anthocyanidins do not undergo any extensive metabolism and thus their bioavailability is quite low. Small amounts of anthocyanins are absorbed in the small intestine with the majority being absorbed in the colon [160]. In a survey of 15 bioavailability studies of anthocyanins, it was found that plasma concentrations were quite low. The only metabolites of anthocyanins were found to be unchanged glycosides, though Phase II metabolites were also present in urine [159]. In studies conducted with anthocyanins, polyphenol ingestion increased brain activation in multiple Brodmann Areas, the precuneus, anterior cingulate, insula, and thalamus. Resting state quantitative perfusion of grey matter in the parietal and occipital lobes also increased and patients had a significantly faster reaction time along with a higher calm rating [161,162]. The effects displayed in the brains of study participants indicates the possibility of anthocyanin or a metabolite acting in the brain. Proanthocyanidins have not been extensively studied, although they are prevalent in quite a few foods. Further investigation is required.

The bioavailability of catechins has been found to be quite variable. EGCG is absorbed quite rapidly with relatively low T_max_ for free EGCG and the area under the curve (AUC) and C_max_ increase proportionally with the dose of total EGCG, with free EGCG making up the majority in the plasma [163]. EGCG is the only polyphenol found in the plasma in its free form as other catechins are highly conjugated from Phase II biotransformations, although galloylated catechins have not been recovered in urine as they are preferentially excreted in bile [159]. It is known that flavonoids are absorbed into the blood and that they are majorly excreted in the urine, though biliary excretion is also possible. Flavonoids are usually metabolized either by glycosylation or acetylation. Glycosylation has a large effect on flavonoid bioavailability by affecting chemical, physiological, and biological properties [77]. With flavonols, monosaccharides are metabolized more quickly than di- or tri-saccharides and flavonoids that are lacking hydroxyl groups at certain carbon positions have slower degradation than those that have them. However, once flavonoids are metabolized to their aglycones, they are degraded by colonic microbiota to generate simple phenolic compounds. Flavanones have a similar path of metabolism to flavonols with the exception of having a slightly different position for C-ring cleavage [150]. Quercetin, a flavonol, is absorbed more quickly from quercetin aglycone than rutin and both are generally present in the plasma as glucuronides and sulfates [164]. Free quercetin is not found in the plasma after dietary interventions [165]. Interestingly, the excretion of quercetin metabolites is quite slow with half-lives between 11 to 28 h [159]. Differing food sources will have different pharmacokinetic properties of polyphenols, and the metabolites of polyphenols can also differ in their absorptive and excretory properties.

## 5. Conclusions

There are several aspects that contribute to the pathology of PD. Currently, there is no known cure for the disease and therefore additional treatments are needed. As summarized in this article, polyphenols have shown benefit in several experimental models of PD and consumption of dietary flavonoids is also associated with lower PD risk in humans. As we understand more about the gut microbiome connection with the brain, we may unravel more information demonstrating the significant impact that diet has on neurodegenerative disorders. Although additional research is needed, there is strong evidence that suggests dietary intake of polyphenols may inhibit neurodegeneration and the progression of PD. Therefore, a diet rich in polyphenols may decrease the symptoms and increase quality of life in PD patients. This also suggests that polyphenolic compounds hold promise to be developed as nutraceutical agents to treat PD and similar diseases.

## Figures and Tables

**Figure 1 molecules-25-04382-f001:**
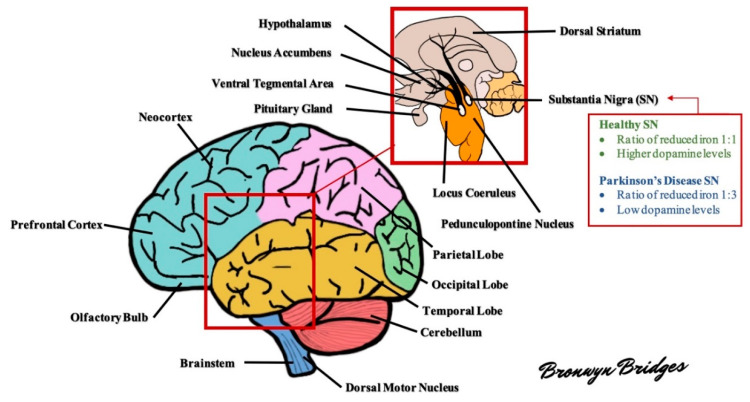
Schematic diagram of the human brain. Several areas of the brain are adversely affected in Parkinson’s disease (PD). For example, the substantia nigra exhibits a profound loss of dopaminergic neurons and altered levels of reduced iron, likely as a result of increased oxidative stress. As the disease progresses, other areas of the brain develop lesions, including the dorsal motor nucleus, neocortex, prefrontal cortex, locus coeruleus, amygdala, and more (see text for further details).

**Figure 2 molecules-25-04382-f002:**
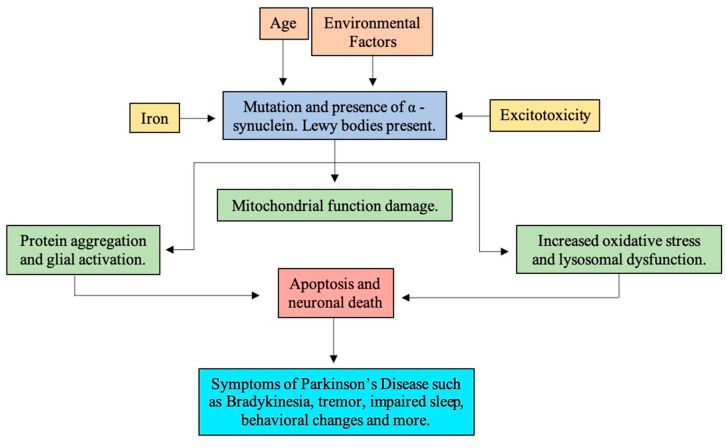
Overview of the pathology of Parkinson’s disease.

**Figure 3 molecules-25-04382-f003:**
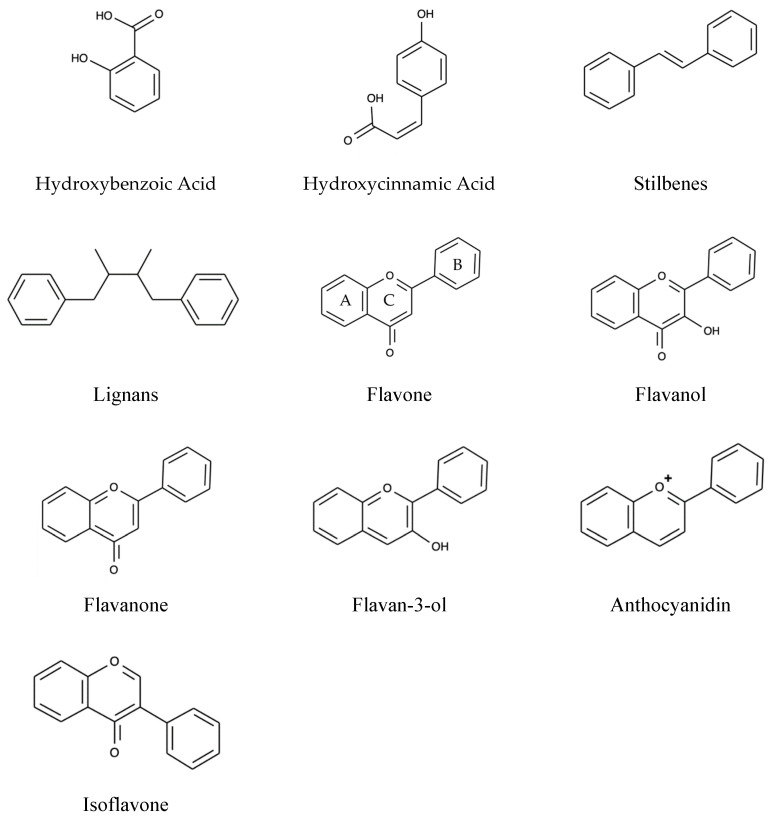
Basic structures of Polyphenols.

**Figure 4 molecules-25-04382-f004:**
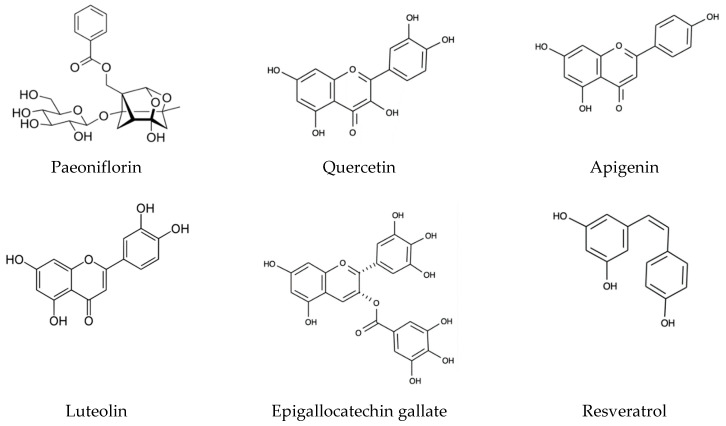
Structures of major dietary polyphenols.

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
