# Peer review of "The Pathology of Parkinson’s Disease and Potential Benefit of Dietary Polyphenols"

_molecules, 2020, doi:10.3390/molecules25194382_

Round 1

Reviewer 1 Report

Authors prepared an interesting article.

Hovewer, I have found some previously published articles, in Scopus base, which are dealing with similar topic(s). If I observed correctly these articles are not cited in current research. So, I suggest to authors to include it in Introduction section.

Here are some examples from Scopus base:

  1. https://doi.org/10.1007/s11064-020-03058-3
  2. https://doi.org/10.1080/14737175.2020.1775585
  3. https://doi.org/10.3390/antiox9070570
  4. https://doi.org/10.3390/biom9070271

All other comments are listed below with an appropriate Line number(s) from text in order to facilitate tracking:

Line 16: This construction "Consumption of dietary flavonoids" seems to me inappropriate since no one will "eat" polyphenols. Suggest to replace with "Intake of polyphenols through diet...".

Lines 40-41: Unclear sentence. Please rewrite.

Lines 42-44: And what is ratio? It is unclear as is written. Define what is UI.

Line 119: Should not be "are reported"?

Line 135: Define abbreviation. It is not used previously.

Line 139: Correct chemical formula.

Line 140: Please make this sign for electron more visible and place it from the right side of formula.

Line 140: Missing something like "during Fenton reaction" or "through Fenton reaction"??

Line 143: Suggest to replace "number five carbon atom" with "C-5 atom".

Line 153: "compound" not "structure".

Line 179: Paraquat is pesticide. So, or merged pesticide and toxins or move it with pesticide.

Line 181: Delete "by". It is surplus here.

Line 192: "Lipids" in plural.

Line 255: Suggest to use Greek letter α. Please unify and apply through a whole Manuscript. For instance in Line 337, 349, 352, etc. it is written as "alpha" but in Line 339 it is given as "α-".

Line 337: Please add Mn, Pb and Hg here since you mentioned it later in text.

Line 383: Without capital letter "L" here.

Line 403: "p-coumaric" - put p in Italic style.

Line 425: Please put "trans" in Italic style.

Line 441: Suggest to specify here that this is cation. After you can use just "MPP+" abbreviation.

Line 457: "polyphenols" not just "phenols".

Line 473: Please check I think that here is one "space" between numbers of cited references but it should be without space between numbers. But it is possible that this underlines make wrong impression. "Potential same error" is repeating in Lines 499, 506, 531, 563, 612, 623, 626, 650 etc. Please check all and correct if it is necessary.

Line 484: Correct this technical error - put charge for Ca ion in superscript.

Line 544: Italic style for Latin name?

Line 550: Give some info about this compound (schisandrin) i.e. plant origin, etc. You did not mentioned before and it is not quite well-known polyphenol.

Line 555: Capital letter C in cytochrome C.

Lines 674-680: I think that all these Latin names should be given in Italic?

Line 699: "The higher the rate of absorption, the higher the rate of secretion" - are you sure that this is not opposite? 

Author Response

We appreciate the reviewers comments on our manuscript. 

We have added the four other review articles mentioned at the beginning of the report. Three of the reviews are cited at the beginning of the second paragraph of section 3. Mechanisms of Neuroprotection and the other reference is cited at the beginning of section 4.1 The Gut Microbiome and Brain Connection.

In addition Please see the Attachment, which has a point by point response to the other comments made by the reviewer. Please note that some line numbers may have changed due to other revisions based on reviewer number 2.

Reviewer 2 Report

Aryal et al. aimed to address here the putative action of dietary flavonoids (also considering microbiome interactions & inflammation) in preventing/controlling the progression of PD.

The review is well-written and particularly accurate on its medical and clinical approach, which highly praised me as a reader and reviewer. The text is also well structured since it presents a logic, intuitive and well concatenated information to address its purposes. 

Regarding oxidative stress, I have major suggestions for the authors to improve the biochemical relevance of their review.

(1) I was particularly glad for the appropriate remarks put on the key participation of iron ions and autooxidation reactons involving cathecolamines (dopamine, L-DOPA etc) in the brain tissue and the PD occurance. Lipid peroxidation in brain tissues is expressive and harmful to trigger PD molecular events when oxidative conditions are not under control. Oxidative stress in brain tissues has been strongly linked with the diffuse distribution/accumulation of iron ions among brain regions - as well as density of dopaminergic receptors (and dopamine production) and antioxidant  activities - which could represent relevant "hotspots" of oxidative injuries associated with PD.

From my point of view, this review significantly lacks illustrations to facilitate its reading, especially when presenting chemical mechanisms. Therefore, I would like to suggest (almost "challenging")  the authors to include a figure that maps the human brain showing PD-relevant brain sections and their respective "iron content", "antioxidant capacity" , and "dopaminergic receptor density", classifying these segments as "high", "medium", or "low". The authors could use the same brain sections presented in the text: SNpc (line 30), dorsal motor nucleus (line 86), neocortex & prefrontal cortex (line 92), other relevant segments (lines 100-105).

(2) Please, include another specific topic focusing on Reactive Nitrogen Species, with special remarks on peroxynitrite (ONOO-) and its oxidative/nitr(osyl)ative reactions. Attention with RNS role in lipid peroxidation.

(3) Somewhere (probably after presenting ROS and RNS), the authors should include, at least, a paragraph describing the reactivity of tyrosyl radical and its relevance in promoting oxidative modifications in biomolecules (focus on neuron lipids, like myelin, and key proteins, etc.)

(4) A whole topic approaching flavonoids and their activation of the Keap1-Nrf2-AREs signaling cascade, leading to cytoprotective gene expression and prevention of PD (or impairing progression) is necessary. The authors could also mention other neurodegenarative diseases, to support this idea, but there are some good references already available about it.

(5) Please, standardize the chemical structures in figures 2 and 3. Same format letters and sizes, bond sizes and line thickness, etc. Use the same software patterns for that.

(6) Again, from my point of view, the key participation of gut microbiome on flavonoid metabolism is a topic of extremely high interest! Therefore, it deserves an extended approach, undoubtedly. I was wondering if this review has space enough for that, specially after describing so well the pathology, clinical aspects, etc of PD and its association with oxidative stress. Therefore, I will rather leave this "open" option/suggestion for the authors:

(i) summarize the introduction topics, including medical aspects and the necessary role of ROS/RNS in PD (adding suggestions above), and EXTEND the chapters about flavonoid metabolism by gut microbiome and redox homeostasis;

OR

(ii) Keep the extended version of redox metabolism in PD as it stands (but please include my suggestions aforementioned) and diminish the remarks on microbiome metabolism, maybe presenting this fascinating topic as "perspectives" .

Please, argue. 

Author Response

We thank the reviewer for their thorough review and overall positive comments on our manuscript. We have addressed as many criticisms as possible in our revised version.

For point 1: we have developed a diagram of the brain as Figure 1, and have pointed out particular areas that are susceptible in PD. However, we found that by including information such as dopamine and iron levels in many areas of the brain it was too complex and difficult to read. We hope that the diagram that we developed will be satisfactory.

For point 2: We have added an additional paragraph on RNS after ROS (section 1.4.1)

Point 3: We have added a paragraph on the tyrosyl radical at the end of the section on dopamine.

Point 4: We have added a paragraph on Keap1-Nrf2-AREs in the section on Flavonoids (section 3.2)

Point 5: All chemical structures have now been standardized.

Point 6: We have more closely addressed the second suggestion here by adding the sections above. We feel that we have not made the manuscript substantially longer, and therefore kept the length of the section on the gut microbiome the same. However, we have added the terms 'Current Perspectives' to this section, which we feel is appropriate since it is a new and excited area with much more to learn.

Overall, we hope that we have addressed the major comments of the reviewer and feel that we have improved the manuscript.